# Hard Wear-Resistant Ti-Si-C Coatings for Cu-Cr Electrical Contacts

**DOI:** 10.3390/ma16030936

**Published:** 2023-01-19

**Authors:** Ph. Kiryukhantsev-Korneev, A. Sytchenko, D. Moskovskikh, K. Kuskov, L. Volkova, M. Poliakov, Y. Pogozhev, S. Yudin, E. Yakushko, A. Nepapushev

**Affiliations:** 1Scientific—Educational Center of SHS, National University of Science and Technology «MISiS», 119049 Moscow, Russia; 2Center of Functional Nano-Ceramics, National University of Science and Technology «MISiS», 119049 Moscow, Russia; 3Department of Physics, Moscow Polytechnic University, 107023 Moscow, Russia; 4Institute of Nanotechnology of Microelectronics of the Russian Academy of Sciences INME RAS, 19991 Moscow, Russia

**Keywords:** Ti-Si-C coatings, magnetron sputtering, electrical contacts, Cu-Cr, tribology, electrical resistance

## Abstract

In this study, hard wear-resistant Ti-Si-C coatings were deposited on Cu-Cr materials to improve their performance as sliding electrical contact materials. A ceramic disk composed of Ti_3_SiC_2_ and TiC phases was used as a target for DC magnetron sputtering to deposit the coatings. The influence of the power supplied to the magnetron on the chemical composition, structure, and friction coefficient of the coatings was examined. The coatings demonstrated high hardness (23–25 GPa), low wear rate (1–3 × 10^−5^ mm^3^/N/m) and electrical resistance (300 μOhm·cm), and fair resistance to electroerosion. The coating deposited at 450 W for 30 min displayed optimal properties for protecting the Cu-Cr alloy from the arc effect.

## 1. Introduction

Sliding electrical contacts can have their service life extended through the use of protective coatings that exhibit high electrical conductivity, low friction, and increased wear resistance [1,2]. Researchers are actively seeking new thin-film coating compositions that meet these requirements as closely as possible, including the use of nanocomposite coatings containing a solid phase dispersed in a softer amorphous matrix. One promising solid phase is titanium carbide (TiC), known for its high hardness (up to 30 GPa) and wear resistance [3,4]. Carbon is a suitable choice for the amorphous matrix, leading to TiC/αC coatings with high hardness (up to 35 GPa), low friction (0.1–0.25), and low specific electrical resistance 5–13 × 10^−4^ Ω·cm [5,6,7,8].

The properties of a nanocomposite coating can be controlled by introducing an additional metal into the carbide phase. This can influence both the microstructure and the stability of the solid carbide phase. Previous research [9,10,11] has demonstrated that non-carbide-forming metals such as Ni and Al can partially replace Ti in the solid TiC phase, leading to the formation of C-C bonds. It is also worth noting that an increase in the αC amorphous matrix content can lead to a decrease in the friction coefficient by 30–50% [12,13]. The properties of nanocomposite coatings can also be modified by changing the composition and volume fraction of the matrix. For example, nanocomposites combining TiC with an amorphous SiC matrix show low electrical resistivity (160–800 μΩ·cm) [14,15] and friction coefficients (0.25–0.40) [16]. By adjusting the concentration of the SiC binder, it is possible to tailor the mechanical properties: coatings with a higher SiC content (30 at.%) have a hardness that is 30% higher than coatings with a lower SiC content (5 at.%) [17]. Ti-Si-C coatings are also promising as protective materials due to their good oxidation resistance at 800–1000 °C [18,19] and record hardness values of up to 40 GPa [14]. Additionally, the presence of the Ti_3_SiC_2_ MAX phase in the coating can enhance its electrical conductivity, as the conductivity of this phase is approximately ~ 15 × 10^3^ S/cm [20]. While the structure, mechanical, and electrical properties of these coatings have been well studied, the phase composition, structure, and performance characteristics of Ti-Si-C coatings obtained by magnetron sputtering of a Ti_3_SiC_2_ MAX phase ceramic target have not been previously explored.

Cu-Cr alloys containing 25–70 wt.% Cr are promising candidates for creating new electrical contact materials for high-voltage electrical circuits [19,20,21,22,23,24]. Recently, a technology was developed for producing nanostructured Cu-Cr pseudo-alloys through the use of the high-energy ball milling (HEBM) and spark plasma sintering (SPS) approaches [22,23], resulting in pore-free solid nanocomposites with specific electrical resistances of 6.0–9.6 μΩ·cm and electrical conductivities >25% IACS [21,24]. One potential way to improve the properties and extend the service life of Cu-Cr electrical contacts is to use coatings in the Ti-Si-C system, which has not yet been explored in previous studies.

The aim of this work is to examine the structure, mechanical, and tribological properties of Ti-Si-C coatings produced through magnetron sputtering using a Ti_3_SiC_2_ MAX phase ceramic target, with the aim of understanding how the electric power supplied to the magnetron affects these properties.

## 2. Materials and Methods

In this study, copper-chromium (Cu-Cr) electrical contact substrates were prepared by mixing commercial powders of Cu (purity 99.97 wt.%, particle size <60 μm) and Cr (purity 99.7 wt.%, particle size 10–30 μm) in a 55 wt.% Cu to 45 wt.% Cr (50:50 vol.%) ratio in a planetary ball mill “Activator-2S” («Chemical Engineering Plant» Ltd., Novosibirsk, Russia). The milling ball (stainless steel, 7 mm diameter) to powder mixture weight ratio was 20:1 (360 g:18 g), at a rotating speed of 694 rpm for the sun wheel and 694 rpm for the jars. It is important to note that 694 rpm is the speed of the rotation of the jars on the sun wheel. HEBM was carried out under an argon atmosphere (4 bar) over 60 min. Bulk Cu-Cr composite was obtained by consolidation into disks of Ø30 mm × 5 mm through spark plasma sintering (SPS) [25,26,27] at a maximum temperature of 900 °C for 10 min. Single-crystal silicon plates (KEF-4.5) were also used as model substrates to study the structure of the coatings. The preparation of the Cu-Cr substrates is described in more detail in [22,23,28].

The coatings were then deposited using DC magnetron sputtering with a Ti-Si-C ceramic disk (Ø120 × 6 mm) as the target, which was produced through the self-propagating high-temperature synthesis (SHS) technology [29,30,31,32] using Ti, Si, and C powders. Prior to the coating deposition, the substrates were cleaned in isopropyl alcohol in a UZDN-2T ultrasonic unit [33] with an operating frequency of 22 kHz for 5 min and etched in a vacuum chamber using Ar^+^ ions (2 keV) for 20 min using an ion source. The coatings were deposited under the following conditions: residual pressure 10^−3^ Pa, working pressure in the vacuum chamber 0.1–0.2 Pa. Three different power regimes were used, as shown in Table 1. To ensure equal thickness, the deposition time was proportionally reduced as the power was increased. Ar with 99.9995% purity was used as the working gas. The magnetron power was controlled through the Pinnacle+ power supply (Advanced Energy, Denver, CO, USA). The elemental composition and structure of the coatings were studied by scanning electron microscopy (SEM) using a JSM-6700F microscope (JEOL, Tokyo, Japan) equipped with a JED-2300F energy dispersive spectrometry (EDS) attachment. The fine structure and chemical composition were studied using a JEOL JEM 2100 Plus transmission electron microscope equipped with a JEOL EX-24261M1G5T energy dispersive analyzer at an accelerating voltage of 200 kV in a bright field. The sample under study was placed perpendicular to the beam direction. Sample preparation was carried out using a Helios G4CX FEI two-beam scanning electron microscope (Thermo Fisher Scientific inc., Waltham, USA). The lamella was treated with a focused ion beam at an accelerating voltage of 30 kV. During the cross section, the current was 0.79 nA, and during polishing it was 24 pA. Before the TEM study, the sample was placed in the FISCHIONE model 1020 Plasma Cleaner, where it was treated in a plasma of 75% argon and 25% oxygen with a high-frequency source for 5 min.

Elemental composition analysis was performed by the glow-discharge optical emission spectroscopy (GDOES) on a Profiler 2 instrument (Horiba Jobin Yvon, Lyon, France) [34]. X-ray diffraction was performed on a D2 Phaser (Bruker, Billerica, MA, USA) diffractometer using CuK_α_ radiation in the range of 2ϴ = 20–80°, using 0.02° steps and an exposure of 0.6 s. Electrical resistivity of the coatings was measured by the four-probe method with a linear arrangement of probes on a VIK-UES apparatus (Measurement Equipment Design & Manufacturing, Moscow, Russia). The mechanical characteristics of the coatings were determined using a Nano-Hardness Tester (CSM Instruments, Peseux, Switzerland) with a load of 4 mN. Tribological tests were conducted on a Tribometer (CSM Instruments) automated friction machine operating according to the “pin-on-disk” scheme. A Cr-Cr rod (6 mm in diameter) with rounded edges was used as a counterbody, and experiments were performed at a 1N normal load and a linear velocity of 10 cm/s. The wear tracks were investigated using a Wyko-1100NT (Veeco, Plainview, TX, USA) optical profilometer. The electrical discharge tests were carried out using the Alier-Metal 303 installation with the following processing parameters: single touch, current of 120 A, voltage of 20 V, pulse duration of 20 μs, and frequency of 640 Hz. A Cu-Cr rod was used as the electrode.

## 3. Results and Discussion

The microstructure and phase composition of the Ti-Si-C target are shown in Figure 1. According to the EDS data, the target had a composition of 55 at.% Ti, 10 at.% Si, and 35 at.% C. Point analysis revealed the presence of regular-shaped TiC grains and elongated grains of the Ti_3_SiC_2_ MAX-phase (Figure 1a). The XRD data showed that the target contained the Ti_3_SiC_2_ MAX-phase and the TiC phase (Figure 1b), which confirms the EDS results. The formation of the MAX-phase in the bulk Ti-Si-C SHS materials was previously observed in [35,36].

Table 1 shows the depth-averaged concentrations of the coatings deposited on Si substrates.

Figure 2 shows typical GDOES data for the obtained materials.

The elements in the coatings were evenly distributed over its thickness, which was ~550 ± 50 nm. The growth rates of the coatings 1, 2, and 3 were ~ 20, 25, and 60 nm/min, respectively. Thus, the dependence of the growth rate on power is linear, similar to the observations in [37]. This increase in the deposition rate with sputtering power can be correlated with an increased flux of metal atoms. An increase in the Ar ion flux within the coating chamber increases the probability of ejecting a higher number of metal ions from the target [38].

The coatings contained a small amount of oxygen (<5 at.%), which likely contaminated the materials through the residual gas or pores in the ceramic target. The concentrations of the main elements in coating 1 were 35.2 at.% Ti, 16.5 at.% Si, and 43.4 at.% C, with an oxygen concentration of 4.9 at.% or less. Coating 2 had evenly distributed main elements, with concentrations of 35.8 at.% Ti, 16.3 at.% Si, 45.2 at.% C, and 2.7 at.% O. In coating 3, there was a smooth decrease in Ti and C concentration and an increase in Si concentration (as seen in Table 1). Coating 3 had the lowest oxygen concentration at 0.8 at.%. A comparison of coatings 1–3 showed that increasing the power did not significantly impact the chemical composition, but did result in a proportional decrease in oxygen impurities (as seen in Table 1). The decrease in oxygen concentration may be linked to an increase in the growth rate of coatings with an increase in sputtering power [37,39]. An increase in sputtering power leads to an increase in the ratio of sputtered target atoms to oxygen atoms. It is worth noting that the titanium content in the coatings was lower than that in the target, which may be due to the high scattering of sputtered Ti atoms [40].

Cross-section SEM images showed that the structure of coatings 1–3 was homogeneous, without pronounced columnar elements (as seen in Figure 3).

No significant difference in the structure of the coatings according to SEM images was found. The surface roughness (Ra) of the coatings increased with the power increase, with values of 304, 355, and 360 nm for coatings 1, 2, and 3, respectively. This trend in roughness increase with power increase has been previously observed in [41] and is believed to be related to an increase in the crystallinity of the coatings. Previous studies [17] have shown that Ti-Si-C coatings produced by reactive magnetron sputtering tend to have large grains, which can negatively impact their mechanical properties. It is worth noting that the roughness of the Cu-Cr substrate was 286 nm, which is closest to the value obtained for coating 1.

Since the coatings were characterized by identical structure and phase composition, the transmission electron microscopy (TEM) results are provided for coating 2 deposited in the medium energy regime (Figure 4a).

The TEM image of the coating, deposited in the medium energy regime, shows a columnar structure with columns ranging in size from 5 to 20 nm (Figure 4a). The selected area electron diffraction (SAED) pattern reveals narrow reflections with interplanar spacings of 0.255, 0.217, 0.155, and 0.132 nm, which are similar to the values of 0.251, 0.216, 0.153, and 0.130 nm for the face-centered cubic TiC phase (JCPDS 89-3828). No amorphous inclusions were detected.

Diffraction patterns of coatings 1–3 revealed peaks from the Si substrate (JCPDS 17-0901) and reflections from the (111), (200), and (220) planes of the fcc TiC phase at positions 2θ = 36.0, 41.4, and 60.4°, respectively. The size of the TiC crystallites in coatings 1, 2, and 3, determined using the Scherrer equation with the (111) line, were 15, 16, and 18 nm, respectively. It has been observed that increasing the sputtering power results in an increase in the kinetic energy of the sputtered atoms and improved crystallinity, as seen in NiO and TiO_x_ coatings [42,43]. This is because the sputtering power increase leads to an increase in the kinetic energy of the sputtered atoms, resulting in improved crystallinity [44,45]. The observed absence of the Ti_3_SiC_2_ MAX phase in the coatings can be attributed to their lower titanium content. This lack of titanium prevents the formation of secondary silicon-containing phases, which are important for the formation of the MAX phase. However, titanium can react with silicon carbide to form titanium carbide and silicide (2Ti + SiC = TiC + TiSi [46]). The MAX phase can also be formed through the reaction of titanium carbide and silicide (2TiC + TiSi = Ti_3_SiC_2_ [46]). Previous research has shown that magnetron-sputtered Ti-Si-C coatings with high titanium and carbon content can contain both the Ti_3_SiC_2_ and TiC phases, as well as the Ti_5_Si_3_ secondary phase [16,47]. The latter phase may also form due to high titanium content through the reaction 8Ti + 3SiC = 3TiC + Ti_5_Si_3_ [46]. Increasing the amount of the MAX phase in the coatings can also be achieved by annealing them at temperatures up to 1250 °C [48].

The nanoindentation results showed that all three coatings had similar hardness values: 23 ± 2 GPa for coating 1, 25 ± 4 GPa for coating 2, and 25 ± 2 GPa for coating 3 (Table 1). The elastic modulus for coatings 1–3 was 245 ± 27, 268 ± 21, and 247 ± 14 GPa, respectively. The substrate hardness was in the range of 1–5 GPa [22,23]. It is worth noting that the indentation depth of the indenter was 30 nm, which is 5–6% of the coating thickness. At such a depth (<10%), the substrate has no effect on the hardness of the coatings, as has been shown in detail in [49,50]. According to literature, the hardness of ion-plasma-generated coatings in the Ti-Si-C system is in the range of 21–27 GPa [14,51], which is comparable to the values measured in this study.

All coatings demonstrated similar values of electrical resistance, ranging from 291 to 302 μΩ·cm, which are similar to those reported for nc-TiC/a-SiC films, which have electrical resistances ranging from 250 to 350 μΩ·cm [52]. A slight decrease in resistance was observed with increasing power, which may be due to a decrease in oxygen impurities’ concentration and an increase in coating density. It is worth noting that Ti-Si-C magnetron-sputtered coatings have electrical resistances of 330–340 μΩ·cm [15,53], which are approximately 10% higher than the values obtained in this study. Ti-Ni-C coatings also have higher electrical resistivity values, up to 400 μΩ·cm [54].

Tribological tests showed that all coatings have relatively low friction coefficients (Figure 5).

The friction coefficient of coating 1 increased from 0.14 to 0.24 over a distance of 0–350 m, with an average value of 0.20. Coating 2 had a stable friction coefficient of 0.24 across the entire distance. Coating 3 showed unstable behavior, with friction coefficients fluctuating between 0.26 and 0.33, with a mean value of 0.28. A trend was observed where the friction coefficient increased with increasing power. This may be attributed to the roughness of the coatings and the different TiC crystalline phase content [55]. When the Cu-Cr ball was tested on a substrate with a similar composition, the friction coefficient increased from 0.4 to 0.9 over a distance of 0–60 m. After this, the test was automatically stopped due to adhesive interaction between the disc surface and the rod of the same composition. Therefore, the application of the Ti-Si-C coating significantly reduces the friction coefficient of the Cu-Cr alloy. According to literature data, magnetron-sputtered Ti-Si-C coatings of a similar composition tested under similar conditions have friction coefficients ranging from 0.25 to 0.60 [16,56].

According to the 3D profiles of the substrate and coatings (Figure 5), the coatings showed minimal wear. It should be noted that the abrasive wear mechanism predominated during sliding friction, similar to what has been observed for TiSiC-based coatings produced in [57,58]. The wear track depth of coating 1 was within the roughness limits, making it difficult to determine the coating’s wear rate. The wear rates of coatings 2 and 3 were 0.9 and 2.9 × 10^−5^ mm^3^/N/m, respectively. In comparison, the Cu-Cr substrate tested under the same conditions had a wear rate of 58 × 10^−5^ mm^3^/N/m. The wear rates of the Cu-Cr counterbody were 1.5, 0.7, 2.5, and 40 × 10^−6^ mm^3^/N/m for coatings 1, 2, 3, and the substrate, respectively. It is worth noting that all coatings increased the Cu-Cr alloy’s wear resistance by an order of magnitude. Coating 2 showed the best wear resistance in terms of the minimum combined wear of the coating and counterbody.

The results of the surface study of the coatings after electroerosion tests in contact with the Cu-Cr electrode are shown in Figure 6.

The average erosion depth values for coatings 1, 2, and 3 were 5.8, 6.9, and 7.2 μm, respectively. The erosion depth of the substrate was 7.1 μm. It is worth noting that coating 1 had the smallest number of craters and the narrowest contact zone of 440 μm. An increase in power resulted in an increase in the number of craters and the width of the contact zone for coatings 2 and 3, to values of 590 and 712 μm, respectively. Many wide craters were observed in the erosion zone for the substrate, with the widest contact zone at 790 μm. Therefore, the application of coatings using regime 1 increases the electroerosive resistance of the Cu-Cr substrate used as a contact material by approximately 20%.

## 4. Conclusions

The application of hard, wear-resistant coatings on a Cu-Cr substrate improved their performance as electrical contact materials. Microstructural analysis showed that TiC and SiC grains are the main structural components of the coatings, contributing to their high hardness, low friction coefficient, and low wear rate. By changing the power supplied to the magnetron, it was possible to increase the size of TiC crystallites by 18% and the roughness by 20%. The growth rate increased from 20 to 69 nm/min with an increase in power from 450 to 1230 W, which is likely due to an increase in the metal atom flux. The friction coefficient also exhibited a dependence on power: the coefficient increased from 0.20 to 0.24 and 0.28 with increasing power from 450 W to 800 W and 1230 W, respectively. The lowest friction coefficient and highest wear resistance were observed for the coating produced at a current of 1 A and a power of 450 W, which had the lowest surface roughness. It is worth noting that all coatings effectively protected the Cu-Cr substrate from wear during friction, while the Cu-Cr/Cu-Cr couple adhesively interacted with each other. Electroerosion tests showed that the resistance of the coatings decreases with increasing power. The smallest erosion damage zones were observed for the coating produced at a power of 450 W.

## Figures and Tables

**Figure 1 materials-16-00936-f001:**
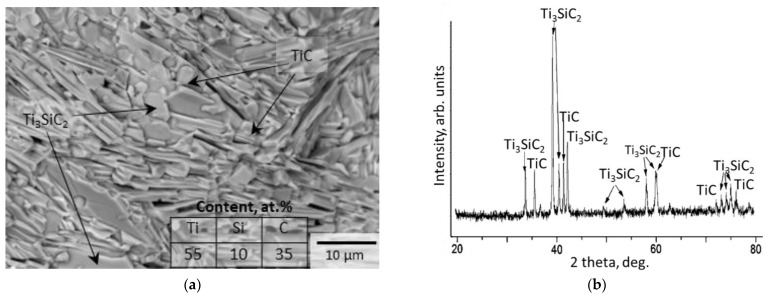
SEM and EDS data (**a**) and diffraction pattern (**b**) of the Ti_3_SiC_2_ target.

**Figure 2 materials-16-00936-f002:**
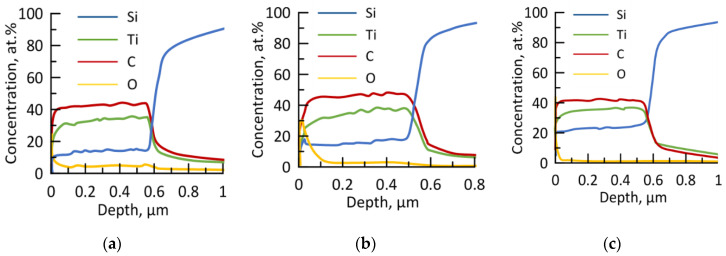
The GDOES profiles of the coatings 1 (**a**), 2 (**b**), and 3 (**c**).

**Figure 3 materials-16-00936-f003:**
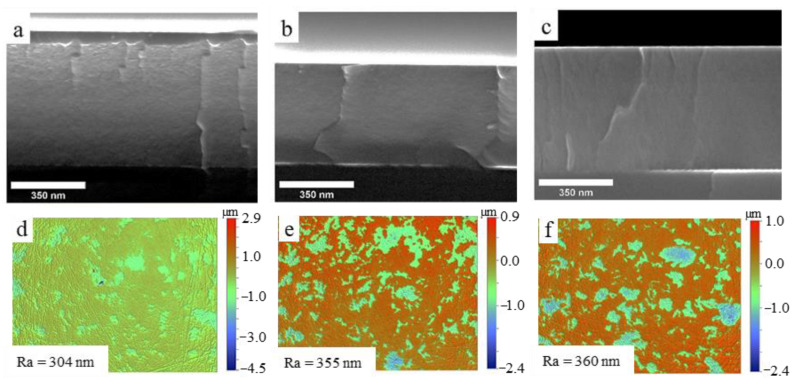
Cross-section SEM and surface images of the coatings 1 (**a**,**d**), 2 (**b**,**e**), and 3 (**c**,**f**).

**Figure 4 materials-16-00936-f004:**
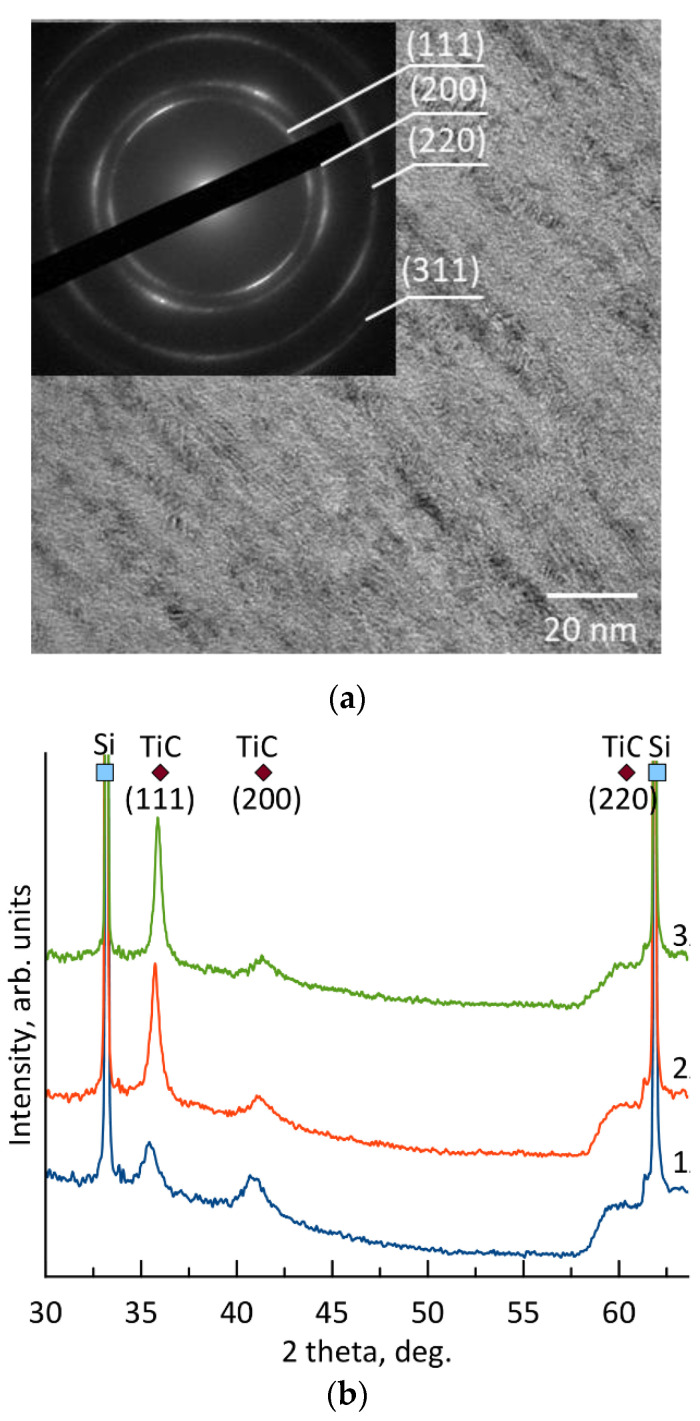
TEM structure image of coating 2 (**a**) and X-ray diffraction patterns (**b**) of coatings 1, 2, and 3.

**Figure 5 materials-16-00936-f005:**
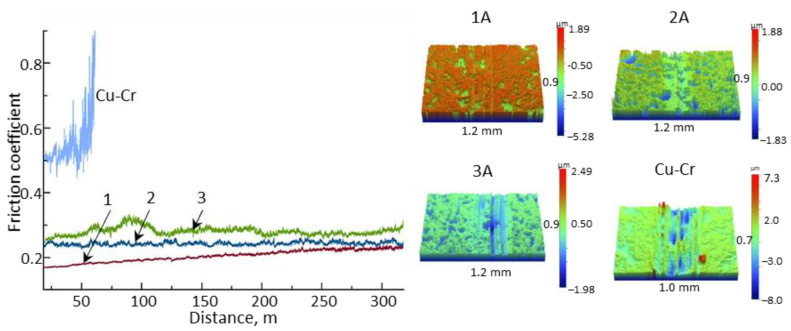
Dependence of the friction coefficient on the distance for coatings 1–3 and uncoated substrate.

**Figure 6 materials-16-00936-f006:**
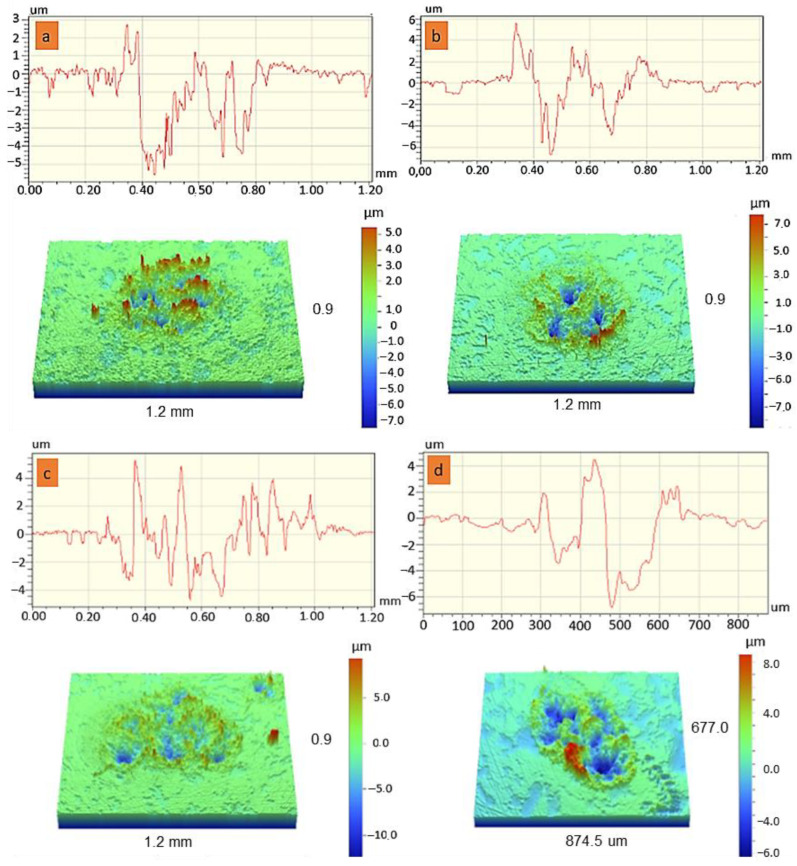
Results of the electroerosion tests for coatings 1 (**a**), 2 (**b**), 3 (**c**), and the Cu-Cr substrate (**d**).

**Table 1 materials-16-00936-t001:** Composition and mechanical properties of the coatings.

№	Current, A	Power, W	Duration, min	Chemical Composition (GDOES), at.%				Ra, nm	H, GPa	E, GPa	Ω, µOhm·cm
				Ti	Si	C	O				
1	1	450	30	35.2	16.5	43.4	4.9	304	23 ± 2	245 ± 27	302
2	2	800	20	35.8	16.3	45.2	2.7	355	25 ± 4	268 ± 21	297
3	3	1230	10	37.1	20.6	41.5	0.8	360	25 ± 2	247 ± 14	291

## Data Availability

Not applicable.

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
