# Peer review of "Hard Wear-Resistant Ti-Si-C Coatings for Cu-Cr Electrical Contacts"

_materials, 2023, doi:10.3390/ma16030936_

Round 1

Reviewer 1 Report

In this work, hard wear-resistant Ti-Si-C coatings were deposited on the Cu-Cr materials to improve their operating characteristics as sliding electrical contacts materials. But there are still some issues to be resolved. Minor revision are recommended.

1.       The authors conclude that resistance properties and roughness cannot be possessed at the same time, so what is the reference significance of the three samples obtained by the authors through modification?

2.       The authors mention that the optimal power is 450 W. Is this the best preparation condition for Ti-Si-C material? Is it possible to reduce the roughness if the power is reduced?

3.       The authors should refine the conclusion part instead of just giving experimental results.

4.       Please improve English grammar and correct some typos in the text.

Author Response

Dear Reviewer,

Thank you for your attention to our article and a positive decision. According to your comments, additional research was carried out and corrections were made to the text of the manuscript.

All additions and changes are shown in the table and highlighted by blue font.

English language improvements are highlighted by yellow.

Reviewer 2 Report

Please provide the parameters for the characterization including SEM, XRD and others.

The font size in the sub-panel should be uniformed. The figure resolution should be improved. It may be better to add some markers to the lines to make the legend clearer.

The biggest problem of this paper is that the optimal processing of manufacturing parameters was not doing well in logic. The authors compared the results by 3 different manufacturing parameters but did not clearly clarify why they chose these. They concluded the 1st recipe to the best, but did not prove it the statistics.

Author Response

(The authors gave the same response as above.)

Reviewer 3 Report

The subject of the paper is interesting and paper could be accepted for the possible publication in Materials Journal, MDPI publication. Nevertheless, the paper requires Minor Revision, prior to the publication. The things that need revision (in order of appearances):

1. The authors are kindly requested to try to improve a few typo error throughout the manuscript.  

2. Add few more recent review articles of the present work  in introduction part.

3.  The authors have made a statement “At the first stage Cu (99.97 wt.%, particle size < 60 μm) and Cr (99.7 67 wt.%, particle size 10 – 30 μm) powders were mixed in a 55 wt.% Cu to 45 wt.% Cr ratios in a planetary ball mill “Activator-2S” (Russia).” Please elaborate on why the above-mentioned weight percentage and size fraction was used for the present study. What is its impact on the results?

4. Why planetary ball mill was used for mechanical treatment instead of regular ball mill. Whether the mechanical treatment involved wet or dry grinding.

5. How grinding will influence the mechanical treatment w.r.t present research work.

6. How the authors have arrived at optimal grinding condition. Please elaborate and specify the optimal grinding condition.

Author Response

(The authors gave the same response as above.)
